# β-Neoendorphin Enhances Wound Healing by Promoting Cell Migration in Keratinocyte

**DOI:** 10.3390/molecules25204640

**Published:** 2020-10-12

**Authors:** Dong Joo Yang, Sang Hyun Moh, Yun-Hee Choi, Ki Woo Kim

**Affiliations:** 1Department of Oral Biology, Yonsei University College of Dentistry, Seoul 03722, Korea; ydj1991@yuhs.ac; 2Department of Global Medical Science, Yonsei University Wonju College of Medicine, Wonju 26426, Korea; 3Anti-aging Research Institute of BIO-FD&C Co. Ltd., Incheon 21990, Korea; biofdnc@gmail.com; 4Department of Internal Medicine, Division of Hypothalamic Research, University of Texas Southwestern Medical Center, Dallas, TX 75390, USA; iamyunhee@gmail.com; 5Department of Applied Biological Science, BK21 FOUR, Yonsei University College of Dentistry, Seoul 03722, Korea

**Keywords:** β-neoendorphin, endogenous opioid, wound healing, human keratinocyte, migration

## Abstract

The skin is the largest and a remarkably plastic organ that serves as a protective barrier against environmental stimuli and injuries throughout life. Skin injuries are serious health problems, and wound healing is a critical process to replace devitalized cellular and tissue structures. Although some endogenous opioids are known to be involved in the modulation of wound healing, it remains to be determined whether the β-neoendorphin (β-NEP), an endogenous opioid, has beneficial effects on wound repair in human keratinocyte. In this study, we found that β-NEP accelerated wound repair through activation of mitogen-activated protein kinase (MAPK)/Erk1/2 signaling pathways in human keratinocytes. Moreover, the wound healing effect of β-NEP is mainly through the acceleration of keratinocyte migration without affecting cell proliferation. Therefore, our studies reveal that β-NEP plays an important role in the regulation of wound repair and suggest a therapeutic strategy to promote wound healing using β-NEP.

## 1. Introduction

The skin is the largest organ in the body and functions as a barrier between the external and internal environments [1]. The skin is continuously exposed to various environmental stresses including physical, chemical, and biological insults. Thus, skin has the capability to locally recognize, discriminate, and integrate various signals within highly heterogeneous environments, and to immediately respond appropriately [1]. The skin is composed of an outer epidermal sheet and an underlying connective tissue or dermis. Multiple cell populations and matrix components form distinct yet interdependent compartments that confer remarkable plasticity on skin. These functions in skin are coordinated by the local neuroendocrine and immune systems that produce biological active compounds such as hormones, neurohormones, neurotransmitters, and cytokines [1,2,3,4,5,6].

Opioid receptors are expressed on peripheral sensory nerve endings, cutaneous cells, and immune cells [7]. The opioid receptor family comprises three members: the μ-, δ-, and κ-opioid receptors, which belong to the G-protein coupled receptor (GPCR) superfamily. The endogenous opioid system includes enkephalins, endorphins, dynorphins, and endomorphins, as well as three opioid receptors. Various cells including immune cells and keratinocytes in the skin can produce opioid peptides [8,9]. These endogenous opioids directly interact with opioid receptors located in immune cells and nociceptive nerve terminals of the skin [8]. Opioid receptor signaling influences cell differentiation and migration as well as cytokine expression in human epidermis. In this regard, the opioid system plays an important role not only in normal skin homeostasis but also in wound healing and scar formation [5,10,11].

Despite the constant exposure to the physical, biochemical, and radiation injury, the skin repairs wounds and maintains homeostasis, which are important for survival [12]. Wound healing is processed as sequential activation of local and systemic cells that function in concert to restore skin integrity. It consists of at least three different phases: the inflammatory phase, re-epithelialization (the proliferation phase), and the maturation phase stimulated by cytokines [13]. Enhanced migration of keratinocytes indicates a positive effect on the healing process, and delayed migration indicates a negative effect. In the final remodeling phase, the reorganization of scar tissue occurs.

A growing body of evidence has shown the stimulating effects of endogenous and exogenous opioids on the migration, formation of granulation, and re-epithelialization in keratinocytes [5,9,14,15,16,17,18]. In oral epithelial keratinocytes, the application of morphine enhanced cell migration and wound closure through δ-opioid receptors [14]. A recent study demonstrated that the opioid peptides released in wound tissue were crucial for re-epithelialization and tissue regeneration [9]. In addition, enkephalin plays a role in the differentiation of epidermal keratinocytes and contributes to the formation of skin protective barrier against pathogenic environment [4,19]. These results signify that understanding the functional roles of opioids in the regulation of wound repair will confer not only molecular insight but also therapeutic strategy for wound healing.

Both α- and β-neoendorphin (β-NEP) are a group of endogenous opioid peptides derived from the proteolytic cleavage of prodynorphin (ProDyn) and bind μ-, δ-, and κ-receptors with distinct affinity [20,21,22]. Recent study showed that opioid receptors are important in skin homeostasis, epidermal nerve fiber regulation and pain modulation. In addition, studies using μ-, δ-, and κ-receptors specific knock-out (KO) animals demonstrated delayed wound closure in KO mice, implying a critical role of the receptors in wound healing [23]. In the current study, we therefore investigated functional roles of β-NEP in the regulation of wound healing in human keratinocytes and sought to explore the molecular mechanism involved in β-NEP-mediated wound repair.

## 2. Results

### 2.1. β-NEP Stimulates Wound Healing in Human Keratinocytes

To investigate the effect of β-NEP and to find optimal dose on wound healing, cell scratch assays in the presence of different doses of β-NEP were performed in human keratinocytes (HaCaT). Wound healing was examined based on the efficiency of monolayer cells invading the wound region following treatment with β-NEP for 0 (T0) or 20 h (T20). β-NEP treatment at a dose of 10 and 20 μM significantly increased the wound closure (Figure 1A,B). In addition, β-NEP treatment ranging from 5 to 20 μM had no observable effect on cell viability (Figure 1C). The 10 μM β-NEP showed a potent wound healing effect comparable to epidermal growth factor (EGF), a known wound healer (Figure 1D,E). Therefore, 10μM of β-NEP was used for all the subsequent experiments. Together, these results indicate that β-NEP stimulates wound closure in keratinocytes.

### 2.2. β-NEP Accelerates Wound Healing through Activation of Ekr1/2

Erk1/2 has been known to be involved in cell proliferation and migration during wound healing [24,25,26]. Thus, HaCaT cells were treated with β-NEP for indicated time periods and Erk1/2 activation was examined. As shown in Figure 2, Erk1/2 was activated by β-NEP treatment (Figure 2A). Previous findings suggested that the Erk1/2 activation and its effector P90^RSK^ phosphorylate β4 integrin and plectin, which is required for hemidesmosome disruption leading to cell migration [27,28]. Moreover, Erk1/2 phosphorylates transcription factors, including Elk-1, which is mainly engaged in cell migration by regulating matrix metalloproteinase (MMP)-2 and -9 expression [29]. As the β-NEP activated Erk1/2, we next examined downstream effectors of Erk1/2. The human keratinocytes treated with β-NEP were subjected to Western blotting for anti-pP90^RSK^ and pElk-1. Both P90^RSK^ and Elk-1 were phosphorylated by β-NEP treatment (Figure 2A,B). To further confirm the roles of Erk1/2 in the regulation of the β-NEP-induced wound healing, HaCaT cells were pretreated with MEK1 inhibitor, PD98059 before EGF and β-NEP stimulation. Phosphorylation of Erk1/2, P90^RSK^, and Elk-1 was significantly suppressed by treatment of the PD98059 (Figure 2B). In addition, the blockage of Erk1/2 activity using PD98059 resulted in a marked reduction of wound healing mediated by β-NEP (Figure 2C,D). Similarly, treatment with U0126, a selective inhibitor of both MEK1 and MEK2, also markedly blunted the wound healing effect of β-NEP (Figure 2E,F). Taken together, these results indicate that β-NEP promotes wound healing through activation of the Erk1/2 signaling pathway in human keratinocytes.

### 2.3. β-NEP Upregulates MMP-2 and -9 Expression in Human Keratinocytes

Matrix metalloproteinases (MMPs) participate in reorganizing the extracellular matrix under physiological and pathological environment and their protease activity is important for cell migration [30]. Recent studies documented that Elk-1 was implicated in the expression of MMP-2 and -9 [31,32]. As the β-NEP activated Elk-1, we investigated whether MMP-2 and -9 were induced by β-NEP in human keratinocyte. Quantitative RT-PCR analysis showed significant upregulation of MMP-2 and -9 expression but not in other genes specifically involved in inflammatory responses (Figure 3A and Appendix A). In addition, gelatin zymography assay also exhibited increased MMP-2 and -9 enzymatic activities (Figure 3B). These results highly suggest that β-NEP might promote the keratinocyte migration by stimulating MMP-2 and -9 activities (Figure 3A,B).

Although activation of MMPs majorly accelerate cell migration, the activation of Erk1/2 signaling can mediate both proliferation and migration. For this reason, we examined whether β-NEP affects cell migration or proliferation or both to promote wound healing in keratinocytes. To address this question, we performed cell proliferation assay. Different from the stimulatory effect of EGF, β-NEP showed no effect on the cell growth (Figure 3C). In addition, β-NEP still promoted wound closure in the presence of mitomycin C (MMC), a cell proliferation blocker, excluding the possibility that β-NEP-mediated wound healing is caused by proliferation (Figure 3D,E). Thus, these results indicate that the wound healing effect of β-NEP in keratinocyte is mediated by the stimulation of cell migration without alteration of proliferation.

### 2.4. β-NEP Stimulates Fibroblast Migration

Besides the keratinocytes, fibroblasts play an essential role in producing and remodeling the extracellular matrix and in the formation of tissue granulation during the wound repair process [33,34]. Therefore, we examined whether β-NEP promotes wound healing in fibroblasts. To this end, we performed wound healing assay using mouse embryonic fibroblasts (MEF). Wound closure was significantly accelerated by β-NEP treatment, indicating the positive effect of β-NEP on fibroblasts wound healing (Figure 4A,B). Since both proliferation and migration contribute to wound repair, it is necessary to establish the major effect of β-NEP on fibroblasts. Thus, we treated the fibroblasts with MMC to block cell proliferation and then the wound healing assay was performed. Treatment of β-NEP markedly increased fibroblast migration, even in the presence of MMC (Figure 4C,D). These results highly indicate that β-NEP promotes migration in fibroblast.

## 3. Discussion

The role of opioid receptor system has been previously described in wound healing. In rat, local application of the opioid peptide dalargin, a Leu-enkephalin analogue, induced the proliferation and growth of capillaries and improved the re-epithelialization of the wound site [35,36]. A recent study also demonstrated the effects of morphine on cell migration, viability, and proliferation in oral epithelial cells [14]. Moreover, we previously observed that leucine-enkephalin promotes wound healing through regulating hemidesmosome and the expression of MMPs [37]. An essential role of μ-, κ-, and δ-opioid receptor in skin homeostasis including differentiation, proliferation, and migration has been investigated using knockout animal [10,23]. All these results together suggest that opioids are considerably involved in skin wound healing. Seven kappa-preferring endogenous peptides generated from ProDyn were identified and have distinct properties on binding affinity [38], tissue distribution, and enzyme sensitivity [39,40,41,42,43], suggesting that these opioid peptides might play different roles in the regulation of physiology. Among seven identified κ-opioid receptor agonists, three peptides including Dyn A, B, and α-NEP are major agonists for κ-opioid receptor. Although β-NEP is generated from same precursor peptide with α-NEP and involved in pain regulation like other κ-opioid receptor agonists, β-NEP has a different affinity for opioid receptors. Moreover, the physiological role of β-NEP in skin wound healing is completely unknown. Therefore, we studied the effect of β-NEP to investigate whether the peptide has any impact on wound healing in human keratinocytes.

In this study, we showed activation of MAPK/Erk1/2 signaling pathway by β-NEP and it contributed to the wound healing process in human keratinocytes (Figure 1 and Figure 2). During the skin wound healing process, keratinocyte migration is orchestrated by numerous growth factors including EGF and cytokines [44,45]. It has been shown that the EGF-mediated cell migration is mediated by Protein kinase C (PKC) and Erk activation, and these activations lead to hemidesmosome disruption required for cell migration [27]. Although we did not monitor whether the PKC signaling is activated by β-NEP, a recent study showed that activation of δ-opioid receptor promotes skin wound healing by PKC activation [46]. Although there was no different expression level of opioid receptors (Appendix A), it would be interesting to examine whether the wound healing effect of β-NEP is mediated, at least in part, by activation of the PKC pathway.

Cell migration and proliferation are essential processes that maintain the integrity of the skin barrier by re-epithelialization of wounded tissue, contributing to wound healing [47]. Keratinocyte migration is a critical step in orchestrated processes of wound healing [48]. Our result indicates that β-NEP improved wound healing mainly by stimulating migration without effects on proliferation in human keratinocytes (Figure 3). On wounding, dramatic rearrangements of cytoplasmic and membrane structure occurred in keratinocytes along the wound edge. These changes include the disassembly of the cell-extracellular matrix (hemidesmosome), the retraction of keratin intermediate filaments, and the rearrangement of actin cytoskeleton, and the events change the overall morphology of keratinocytes to start the polarized movement [49]. These dynamic rearrangements of cytoplasmic and membrane structures together with β-NEP receptor dynamics might induce diverse migratory patterns in the keratinocytes. Therefore, a future study identifying a critical healing phase in which the β-NEP and β-NEP receptor has a positive effect would be interesting to understand a specific β-NEP role in wound healing process.

Wound healing requires the coordinated process with different types of cells, including keratinocytes, fibroblasts, endothelial cells, macrophages, and platelets, which lead to cell proliferation, migration, and remodeling [50]. Fibroblasts are important for the remodeling of dermal structure and the formation of granulation tissue. Fibroblast can be differentiated into myofibroblast by the stimulation of several factors, including growth factors and the myofibroblast together with fibroblast secrete extracellular matrix components and collagen. Therefore, it would be an intriguing future study to investigate whether β-NEP plays a functional role in transition from fibroblast to myofibroblast. In addition, specific signaling pathways governing the secretion of extracellular matrix protein and collagen either by fibroblast or myofibroblast has not been fully understood. In this regard, research focused on revealing a molecular mechanism that explains how fibroblast and myofibroblast regulate the required matrix protein for wound closure would be necessary.

The stimulating effect of β-NEP on fibroblast migration implies that β-NEP might be a good candidate for wound care agent targeting not only for keratinocyte but also for fibroblast. Thus, our current study suggests that the endogenous opioid peptide, β-NEP, would be an excellent candidate for a potent wound healer through the activation of mitogen-activated protein kinase (MAPK)/Erk1/2 signaling pathways.

## 4. Materials and Methods

### 4.1. Cell Culture and Reagent

Human keratinocytes (HaCaT) were obtained from Dr. Fusenig (Deutsches Krebsforschungszentrum, Heidelberg, Germany), and mouse embryonic fibroblasts (MEF) were obtained from Dr. Benoit Viollet (Institute Cochin, INSERM, University of Paris Descartes, France). The cells were grown in Dulbecco’s Eagle’s medium (DMEM, Low glucose, Hyclone, Logan, UT, USA) containing 10% heat inactivated fetal bovine serum (FBS) and 1% penicillin-streptomycin at 37 °C in 5% CO_2_. Epidermal growth factor (EGF, 30 ng/mL) was purchased from Sigma-Aldrich (St. Louis, MO, USA). PD98059 (an ERK inhibitor, 50 μM) and U0126 (MEK inhibitor, 10 μM) were purchased from Cell Signaling (Danvers, MA, USA). Mitomycin C, cell proliferation inhibitor, was obtained from Abcam (Cambridge, UK). β-NEP was synthesized and provided by BIO-FD&C, Ltd. (Incheon, Korea, http://www.biofdnc.com/main/main.html).

### 4.2. MTT Assay

Cell viability was determined by the 3-(4,5-dimethylthiazol-2-yl)-2,5-diphenyltetrazolium bromide (MTT, Sigma-Aldrich, St. Louis, MO, USA) assay. HaCaT cells were cultured at a concentration of 2.5 × 10^4^ per well in 96-well microplates and incubated in media containing 10% FBS. A total of 24 h later, cells were treated with the indicated doses of β-NEP for 24 h and subjected to MTT assay. After treatment, the media was replaced with 100 μL of MTT solution (0.5 mg/mL in cell culture media) and incubated at 37 °C for 4 h. The MTT formazan dissolved in 100 μL dimethyl sulfoxide (DMSO) after removing the incubated solution. Absorbance was measured at 540 nm using spectrophotometry. The results were expressed as mean ±SEM of three independent experiments.

### 4.3. Wound Healing and Migration Assays

Cells were cultured in six-well plates and grown to confluence at 37 °C in 5% CO_2_. A scratch wound was created by scraping the cell monolayer with a P100 pipet tip. Cells were washed once and incubated with 2 mL of growth media containing EGF, β-NEP, or vehicle.

For migration assay, HaCaT and MEF cells were cultured in culture media. After 24 h, the culture media was aspirated and replaced with media containing 0.5% FBS and 1% penicillin-streptomycin. A scratch wound was created using a P100 pipet tip and the cells were treated with mitomycin C (2.5 μg/mL for HaCaT, and 5 μg/mL for MEF cells). Images were acquired immediately using an AxioObserver FL microscope (Advanced Microscopy Group, Bothell, WA, USA) at ×10 magnification. The width of the wound and migration distances were measured in more than three locations, and the percentage wound healing area and migration distances were calculated using ImageJ software (ImageJ 1.52a_Java 1.8.0_112) (http://imagej.nih.gov/ij/index.html). Each experiment was performed in triplicate.

### 4.4. Cell Proliferation Assays

Cell proliferation was determined using the CellTiter 96^®^ Aqueous One Solution Reagent (MTS, Promega, Madison, WI, USA) assay. HaCaT cells were seeded at a concentration of 5 × 10^4^ per well in 96-well microplates and incubated in media containing 10% FBS. A total of 24 h later, HaCaT cells were treated with the indicated doses of β-NEP, EGF, vehicle, and mitomycin C (2.5 μg/mL), and subjected to MTS reagent 2 h before recording absorbance. Absorbance was measured at 490 nm using spectrophotometry. The data are expressed as mean ± SEM of three independent experiment.

### 4.5. Western Blot Analysis

Cells were washed with phosphate buffered saline (PBS) 1× and lysed with Radioimmunoprecipitation assay (RIPA) buffer [150 mM NaCl, 50 nM Tris, 1% Triton-X-100, 0.5% sodium deoxycholate and 0.1% sodium dodecyl sulfate (SDS)] containing protease and phosphatase inhibitors (Roche, Basel, Switzerland). Extracts were isolated by centrifugation at 12,000× *g* for 1 min. The protein concentration was measured using Bio-Rad Protein Assay reagent (Bio-Rad Laboratories, Hercules, CA, USA). Equal amounts (30 μg per well) of protein were loaded and then transferred to nitrocellulose membranes, and electrophoresis was done on SDS—polyacrylamide gels. The membranes were blocked in 5% skim milk in Tris-buffered saline containing 0.1% Tween 20 for 1 h at room temperature. The blocked membranes were then incubated with primary antibodies overnight at 4 °C with agitation followed by incubation with horseradish peroxidase-conjugated secondary antibodies for 1 h at room temperature. The blots were visualized using the Chemiluminescence Western Blot Detection System (BioSpectrum^®^600 Imaging System, Ultra-Violet Products Ltd. Cambridge, UK). Primary antibodies used were as follows: pErk1/2, Erk1/2, pP90^RSK^, pElk-1 (Cell Signaling, Danvers, MA, USA), GAPDH (Santa Cruz Biotechnology, Santa Cruz, CA, USA) and Beta-Actin (GeneTex, Irvine, CA, USA).

### 4.6. Quantitative Real-time PCR

Total RNA was isolated from cells using TRIzol reagent according to the manufacturer’s protocol (Invitrogen, Carlsbad, CA, USA). For quantitative real-time PCR, cDNAs were synthesized using a High-Capacity cDNA Reverse Transcription Kit (Applied Biosystems, Foster City, CA, USA) and real-time PCR was performed in triplicate using the 7900HT Fast Real-Time PCR System (Applied Biosystems, Foster City, CA, USA). *18S* was used as the control gene for normalization. Specific primers for human used were as follows:

*MMP-2*; Forward (F) 5′-CCACTGCCTTCGATACAC-3′, Reverse (R) 5′-GAGCCACTCTCTGGAATCTTAAA-3′, *MMP-9*; F 5′-TTGACAGCGACAAGAAGTGG-3′, R 5′-GCCATTCACGTCGTCCTTAT-3′, *STAT3*; F 5′-AGAGGCGGCAACAGATTGC-3′, R 5′-TTGTTGACGGGTCTGAAGTT-3′, *IL1A*; F 5′-CTTCTGGGAAACTCACGGCA-3′, R 5′-AGCACACCCAGTAGTCTTGC-3′, *IL6*; F 5′-AGACAGCCACTCACCTCTTCAG-3′, R 5′- TTCTGCCAGTGCCTCTTTGCTG-3′, *IL10*; F 5′-GGTCCTCCTGACTGGGGTGAG-3′, R 5′-CGGAGATCTCGAAGCATGTT-3′, *IL22*; F 5′-AGCCCTATATCACCAACCGC-3′, R 5′-TCTCCCCAATGAGACGAACG-3′, *OPRM1*; F 5′-TGACGCTCCTCTCTGTCTCA-3′, R 5′- CCGAGACTTTTCGGGTTCCA-3′, *OPRD1*; F 5′-GCCCATCCACATCTTCGTCA-3′, R 5′-TCGAGGAAAGCGTAGAGCAC-3′, *OPRK1*; F 5′-CGTGATCATCCGATACACAAAGA-3′, R 5′-GACCGTACTCTGAAAGGGCA-3′, and *18S*: F 5′-AACCCGTTGAACCCCATT-3′, R 5′-CCATCCAATCGGTAGTAGCG-3′.

### 4.7. Gelatin Zymography Assay

For the analysis of MMP activity, cells were treated either with vehicle or β-NEP for 12 h. Then, protein samples were prepared [51] and loaded without heating on a 10% SDS-polyacrylamide gel including 2 mg/mL gelatin from porcine skin (Sigma-Aldrich, St. Louis, MO, USA). After electrophoresis, the gels were washed in 2.5% Triton X-100 for 30 min at room temperature to allow the proteins to renature and then incubated at 37 °C overnight in the substrate buffer (50 mmol/L Tris-HCl, 200 mmol/L NaCl, 10 mmol/L CaCl_2_, 1 μmol/L ZnCl_2_). Proteins were stained with Coomassie Brilliant Blue R-250 solution (Sigma-Aldrich) to reveal zones of lysis, and band intensity was calculated using ImageJ software.

### 4.8. Statistics

The data are represented as mean ±SEM. Statistical significance was determined by a 2-tailed Student’s *t*-test and a one-way analysis of variance (ANOVA) Friedman’s test. GraphPad PRISM version 5.0 was used for the statistical analyses, and * *p* < 0.05 was considered as a statistically significant difference.

## Figures and Tables

**Figure 1 molecules-25-04640-f001:**
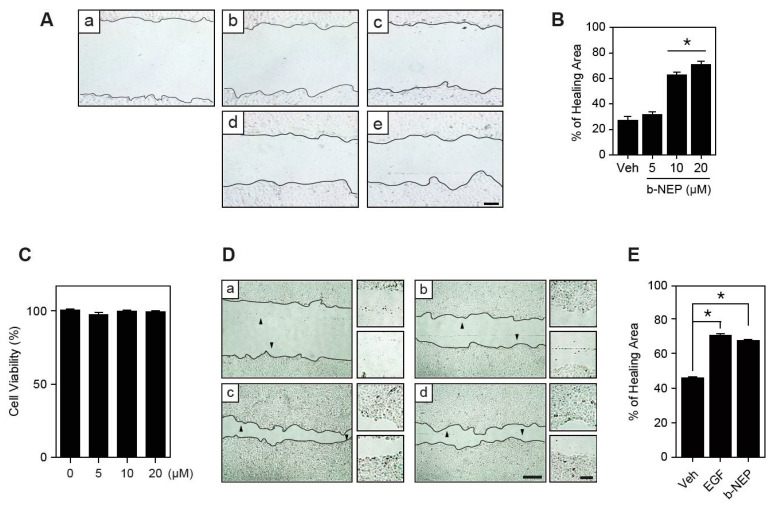
β-neoendorphin (β-NEP) promotes wound healing in human keratinocyte. (**A**) Dose-dependent wound healing effect of β-NEP. Human keratinocyte (HaCaT) cells were grown to confluence and scratch wound was created using pipet tips (a, T0). Cells were incubated for 20 h in the absence (b, Veh) or presence of 5, 10, 20 μM of β-NEP, respectively (c–e). Scale bar = 100 μm. (**B**) The wound area was measured and percent wound healing calculated using ImageJ software. (**C**) Cell viability was assessed by MTT assay after β-NEP treatment for 24 h. (**D**) HaCaT cells were grown to confluence and scratch wound was created using pipet tips (a, T0). Cells were incubated for 20 h in the absence (b, Veh) or presence of epidermal growth factor (EGF) (30 ng/mL, c) and β-NEP (10 μM, d). Scale bar = 200 μm. The arrowhead indicates that the region of magnification and magnified cell image are located on the right (scale bar = 20 μm). (**E**) Percent wound healing from (**D**). Photomicrographs were acquired using an AxioObserver FL microscope at 10× magnification. Results represent the means of three independent experiments. Data represent mean value ±SEM (* *p* < 0.05, one-way ANOVA Friedman’s test).

**Figure 2 molecules-25-04640-f002:**
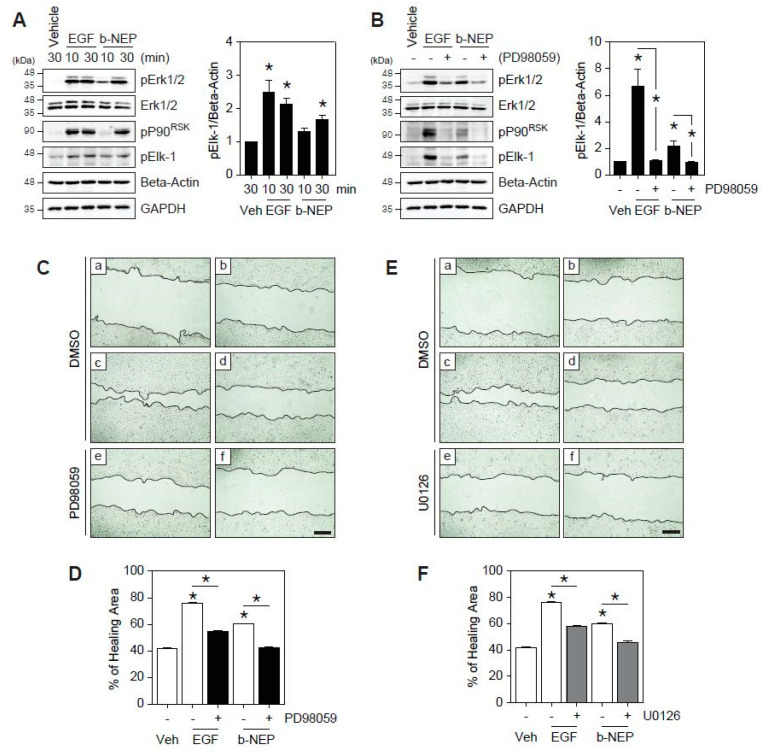
β-NEP regulates cell migration through mitogen-activated protein kinase (MAPK)/Erk1/2 activation during wound healing. (**A**) HaCaT cells were treated with EGF, β-NEP, or vehicle for indicated time periods. Cells were subjected to Western blotting with anti-Erk1/2, phospho-Erk1/2, phospho-P90^RSK^, and phospho-Elk-1antibodies. The graph on the right shows the intensity of phospho-Elk-1 normalized to anti-Beta-Actin (* *p* < 0.05, Student’s *t*-test). (**B**) Cells were treated with EGF, β-NEP or vehicle in the presence or absence of PD98059 (50 μM), an inhibitor of MEK1. The graph on the right indicates the intensity of phospho-Elk-1 normalized by anti-Beta-Actin. (**C**) Confluent monolayers of HaCaT cells were pretreated with dimethyl sulfoxide (DMSO) (a, b, c and d) or PD98059 (50μM, e, f) at 30 min before scratching. Scratch wound was created using pipet tips (a, T0), and cells were treated with EGF (c, e) or β-NEP (d, f) and incubated for 24 h. (**D**) Healing area (%) from (**C**). (**E**) HaCaT cells were pretreated with DMSO (a, b, c and d) or U0126 (10μM, e, f) at 30 min before scratching, and treated with EGF (c, e) or β-NEP (d, f) for 24 h. T0 (a) is the image taken right after scratching. (**F**) Healing area (%) from (**E**). The results represent the mean of three independent experiments. Data represent mean value ±SEM (* *p* < 0.05, One-way ANOVA Friedman’s test). Scale bar = 200 μm.

**Figure 3 molecules-25-04640-f003:**
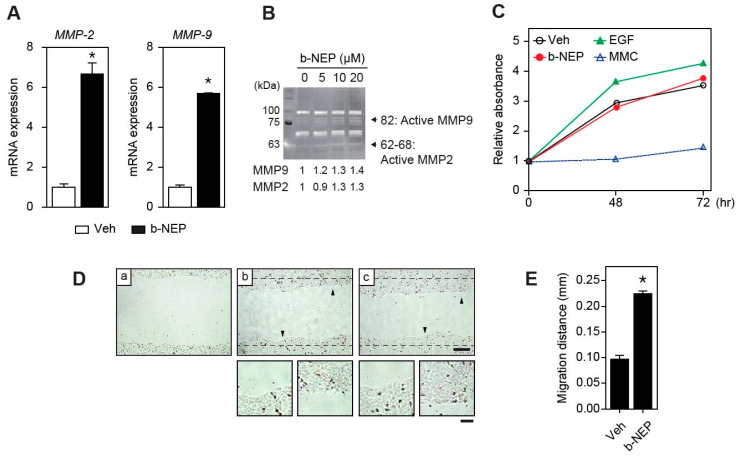
β-NEP stimulates keratinocyte migration through activation of metalloproteinase (MMP)-2 and -9. (**A**) Expression of MMP-2 and -9 after β-NEP treatment for 24 h. (**B**) Gelatin zymography assay to measure MMP-2 and -9 activities after β-NEP treatment for 12 h. Relative intensity listed below was calculated using ImageJ software. (**C**) Cell growth analysis after β-NEP and EGF treatments for indicated time periods. Mitomycin C (MMC, 2.5 μg/mL) was used to inhibit cell proliferation. (**D**) HaCaT cells were scratched (a, T0) and treated with vehicle (b) or β-NEP (c) for 30 h in the presence of MMC. Scale bar = 200 μm. The arrowhead indicates that the region of magnification and magnified cell image are located at the bottom (scale bar = 20 μm). (**E**) Migration distance was measured from (**D**). Results represent the mean of three independent experiments. Data represent mean value ± SEM (* *p* < 0.05, Student’s *t*-test).

**Figure 4 molecules-25-04640-f004:**
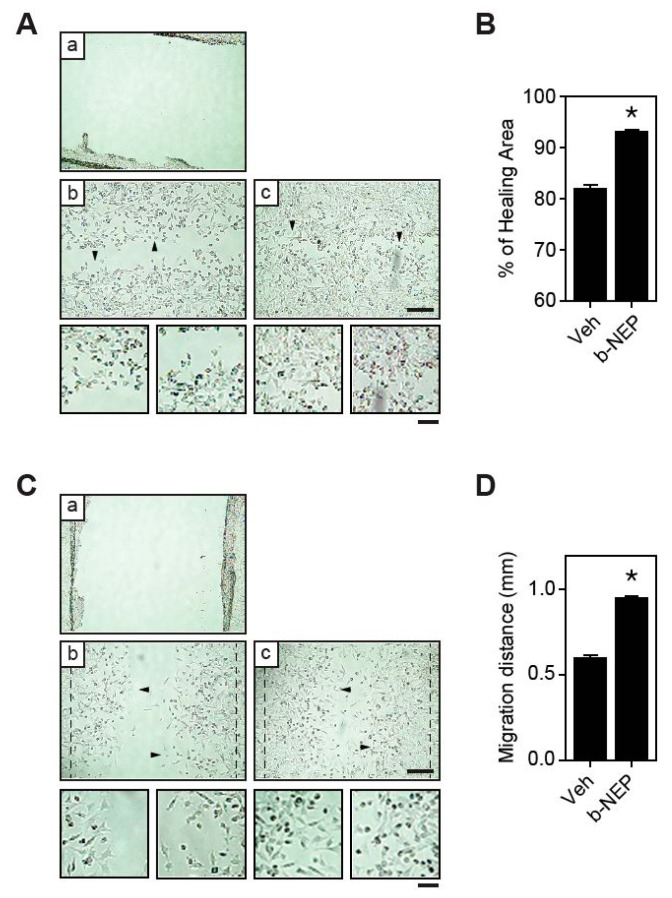
β-NEP stimulates migration in the fibroblast. (**A**) Mouse embryonic fibroblast (MEF) cells were grown to confluence and scratch wound was created using pipet tips (a). Cells were incubated for 12 h in the absence (b) or presence of β-NEP (c). Scale bar = 200 μm. (**B**) Healing area (%) from (**A**). (**C**) MEF cells were scratched (a, T0) and treated with vehicle (b) or β-NEP (c) for 12 h in the presence of MMC (5 μg/mL). Scale bar = 200 μm. (**D**) Migration distance was calculated from (**C**). The arrowhead indicates that the region of magnification and magnified cell image are located at the bottom (scale bar = 20 μm). The results represent the mean of three independent experiments. Data represent mean value ±SEM (* *p* < 0.05, Student’s *t*-test).

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
