# Peer review of "β-Neoendorphin Enhances Wound Healing by Promoting Cell Migration in Keratinocyte"

_molecules, 2020, doi:10.3390/molecules25204640_

Round 1

Reviewer 1 Report

1. Rows 134-135. The authors claim: ”In addition, gelatin zymography assay also exhibited increased MMP-2 and -9 enzymatic activities (Figure S1D).” However, there is a great discrepancy between results of mRNA expression for MMP-2 and MMP-9 (Figure 3A) and the results, depicted in Figure S1D, showing that enzyme activities insignificantly changed. How could the authors argue their claim?

2. Figure 4 A(a). The image is not suggestive at all for what the authors claim. No limit of the scratch is observed, no cell migration front to monitor the event. I suggest to replace this image with a more useful one. Even the migration front is not totally continuous (the cells are fibroblasts) it is better to be seen in the image.

3. MTT Assay. Please specify the seeding density of the cells per well in 96-well microplate.

4. Cell Proliferation Assays. Please specify the seeding density of the cells per well in 96-well microplate.

5. Western Blot Analysis. Please specify the amount of protein loaded on each electrophoresis lane.

Author Response

Response Letter for Reviewer 1

Thank you for the careful reading and comments of our manuscript. We have incorporated the requested changes, and modified the manuscript. Please see the attachment for details.

  1. Rows 134-135. The authors claim: “In addition, gelatin zymography assay also exhibited increased MMP-2 and -9 enzymatic activities (Figure S1D).” However, there is a great discrepancy between results of mRNA expression for MMP-2 and MMP-9 (Figure 3A) and the results, depicted in Figure S1D, showing that enzyme activities insignificantly changed. How could the authors argue their claim?

Response: We agree with the review’s view that there is a discrepancy between mRNA and enzymatic activities of MMP-2 and -9. It is difficult to explain why such a discrepancy exists, but we think that it may be come from the difference in time of β-NEP treatment. For mRNA levels, we incubated β-NEP for 24 hours, but for zymography, we incubated it for 12 hours. To make our data clearer, we updated out manuscript and incorporated β-NEP incubation time for zymography in the Material and Methods section.

  1. Figure 4 A(a). The image is not suggestive at all for what the authors claim. No limit of the scratch is observed, no cell migration front to monitor the event. I suggest to replace this image with a more useful one. Even the migration front is not totally continuous (the cells are fibroblasts) it is better to be seen in the image.

Response: As the reviewer suggested, we have replaced the T0 figure with Figure 4A(a).

  1. MTT Assay. Please specify the seeding density of the cells per well in 96-well microplate.

Response: We apologize for the missing information. We have included all the information in the Materials and Methods section. We seeded cells at a concentration of 5x104 cells/well in 200μl culture media.

  1. Cell Proliferation Assays. Please specify the seeding density of the cells per well in 96-well microplate.

Response: We apologize for the missing information. We have included all the information in the Materials and Methods section. For the cell proliferation assay, we used 2.5x104 cells/well in 200μl in 96-well plate.

  1. Western Blot Analysis. Please specify the amount of protein loaded on each electrophoresis lane.

Response: We apologize for the missing information. We have included all the information in the Materials and Methods section. We determined the total protein concentration of each sample by using the Bradford assay. Then, each sample were load with 30μg per lane.

Reviewer 2 Report

The current study investigated whether the β-neoendorphin (β-NEP), an endogenous opioid, has beneficial effects on wound repair in human keratinocyte. Authors found that β-NEP accelerated human keratinocytes migration through activation of mitogen-activated protein kinase (MAPK)/Erk1/2 signaling pathways and β-NEP did not affect cell proliferation. They also found β-NEP had similar effects on fibroblasts. Here are my concern and suggestions:

1: Are there any data showing the presence and levels of β-NEP in skin wounds either in animal or human studies?

2: All migration figures (Fig. 1-4) should show an image immediately after scratching as a control (0 hour).

3: Fig 4: It seems to have nothing in Fig4Aa. In addition, the cell density in Fig 4 b&c seemed very different which may affect the results.

4: Fibroblast result is a distraction here, suggest to move it to supplement.

5: Move the current supplemental data to the main article.

5: It would be more interesting to show if there are any β-neoendorphin receptors changes after β-neoendorphin treatment.

Author Response

Response Letter for Reviewer 2

Thank you for the careful reading and comments of our manuscript. We have incorporated the requested changes, and modified the manuscript. Please see the attachment for details.

The current study investigated whether the β-neoendorphin (β-NEP), an endogenous opioid, has beneficial effects on wound repair in human keratinocyte. Authors found that β-NEP accelerated human keratinocytes migration through activation of mitogen-activated protein kinase (MAPK)/Erk1/2 signaling pathways and β-NEP did not affect cell proliferation. They also found β-NEP had similar effects on fibroblasts. Here are my concern and suggestions:

  1. Are there any data showing the presence and levels of β-NEP in skin wounds either in animal or human studies?

Response: In our knowledge, there is no report showing levels of β-NEP in skin wounds.

  1. All migration figures (Fig. 1-4) should show an image immediately after scratching as a control (0 hour).

Response: As per the reviewer’s suggestion, we updated all the images for control (0 hour). The first panel, indicated (a) in each figure, is the control, and we also updated the figure legend to clearly indicate the control.

  1. Fig 4: It seems to have nothing in Fig4Aa. In addition, the cell density in Fig 4 b&c seemed very different which may affect the results.

Response: Fig4Aa is the control (T0), as it is the picture taken right after scratching. Fig4Ab and Fig4Ac show the effect of vehicle and β-NEP after 24 hours, respectively, suggesting that the β-NEP has a potent migratory effect in fibroblast.

  1. Fibroblast result is a distraction here, suggest to move it to supplement.

Response: Together with the keratinocytes, fibroblasts play an essential role in producing and remodeling of extracellular matrix and in the formation of tissue granulation during wound repair process. Therefore, wound healing effect of β-NEP in fibroblasts would be important information. Although the reviewer suggested to move the fibroblast result to the supplementary data, all authors would prefer to leave it in its original place. We would like to adhere to the logical flow that the β-NEP has a wound healing effect not only on keratinocytes, but also on fibroblasts.

  1. Move the current supplemental data to the main article.

Response: As the reviewer suggested, we moved the supplemental data to the main figure and updated the Results section.

  1. It would be more interesting to show if there are any β-neoendorphin receptors changes after β-neoendorphin treatment.

Response: We sincerely thank the reviewer for this insightful comment. However, we apologize that we do not have enough time to perform the experiment because of time limitation (10 days rebuttal period were granted) of revision. Therefore, we updated our discussion to include the possible involvement of β-NEP receptor dynamics in the wound healing process.

Round 2

Reviewer 2 Report

I appreciate the authors have make some effort to satisfy my concerns and questions. However, I still some concerns:

1:I think the authors can ask for more time to conduct the experiment I asked in comment 6  "It would be more interesting to show if there are any β-neoendorphin receptors changes after βneoendorphin treatment. "

2: The image in Fig4Aa is still not good enough. It barely shows any cells or wound edges. 

3: There were two scale bars in the figures. All were labeled as 20 or 200um, please indicate which one is 20um, and which one is 200um.  

Author Response

Response Letter for Reviewer 2

Thank you for the careful reading and comments of our manuscript. We have incorporated the requested changes, and modified the manuscript. Please see the attachment for details.

1. I think the authors can ask for more time to conduct the experiment I asked in comment6 “It would be more interesting to show if there are any β-neoendorphin receptors changes after β-neoendorphin treatment.”

Response: We agree with the reviewer’s kind comment. We have performed qPCR analysis for β-neoendorphin receptors (μ-, κ-, δ-opioid receptor) after treatment of β-NEP in human keratinocytes (HaCaT). Briefly, the expression of OPRM1 (μ-opioid receptor), OPRD1 (δ-opioid receptor), and OPRK1 (κ-opioid receptor) did not change after β-NEP treatment. To reflect these results, we provided the qPCR results in Supplementary figure 2 and updated our manuscript.

2. The image in Fig4Aa is still not good enough. It barely shows any cells or wound edges.

Response: As the reviewer suggested, we did our best to replace for better quality of Fig4Aa.

3. There were two scale bars in the figures. All were labeled as 20 or 200um, please indicate which one is 20um, and which one is 200um.

Response: Sorry for the confusion. We have updated all figure legend to make clear for scale bar.

Again, we thank you for careful reading of our manuscript.
